# Rectal Cancer after Prostate Radiation: A Complex and Controversial Disease

**DOI:** 10.3390/cancers15082214

**Published:** 2023-04-09

**Authors:** Dana M. Omer, Hannah M. Thompson, Floris S. Verheij, Jonathan B. Yuval, Roni Rosen, Nathalie R. A. Beets, Anisha Luthra, Paul B. Romesser, Philip B. Paty, Julio Garcia-Aguilar, Francisco Sanchez-Vega

**Affiliations:** 1Department of Surgery, Memorial Sloan Kettering Cancer Center, New York, NY 10065, USA; omerd@mskcc.org (D.M.O.);; 2Department of Epidemiology and Biostatistics, Memorial Sloan Kettering Cancer Center, New York, NY 10065, USA; 3Department of Radiation Oncology, Memorial Sloan Kettering Cancer Center, New York, NY 10065, USA

**Keywords:** radiation, rectal cancer, radiation-associated rectal cancer, colorectal surgery, oncology, surgical oncology, radiation oncology

## Abstract

**Simple Summary:**

Radiation therapy is commonly used in the treatment of prostate cancer, but it is a carcinogen itself. Long-term survivors of prostate cancer who were treated with pelvic radiotherapy have been reported to be at increased risk for developing rectal cancer compared to those treated with surgery. The treatment of radiation-associated rectal cancer is challenging, and the evidence behind this disease is limited and often contradictory. Here, we review the available literature and discuss the unique considerations of treating patients with rectal cancer after prostate radiotherapy.

**Abstract:**

A small proportion of rectal adenocarcinomas develop in patients many years after the treatment of a previous cancer using pelvic radiation, and the incidence of these rectal cancers depends on the length of follow-up from the end of radiotherapy. The risk of radiation-associated rectal cancer (RARC) is higher in patients treated with prostate external beam radiotherapy than it is in patients treated with brachytherapy. The molecular features of RARC have not been fully investigated, and survival is lower compared to non-irradiated rectal cancer patients. Ultimately, it is unclear whether the worse outcomes are related to differences in patient characteristics, treatment-related factors, or tumor biology. Radiation is widely used in the management of rectal adenocarcinoma; however, pelvic re-irradiation of RARC is challenging and carries a higher risk of treatment complications. Although RARC can develop in patients treated for a variety of malignancies, it is most common in patients treated for prostate cancer. This study will review the incidence, molecular characteristics, clinical course, and treatment outcomes of rectal adenocarcinoma in patients previously treated with radiation for prostate cancer. For clarity, we will distinguish between rectal cancer not associated with prostate cancer (RCNAPC), rectal cancer in non-irradiated prostate cancer patients (RCNRPC), and rectal cancer in irradiated prostate cancer patients (RCRPC). RARC represents a unique but understudied subset of rectal cancer, and thus requires a more comprehensive investigation in order to improve its treatment and prognosis.

## 1. The Limitations of Studies Investigating Radiation-Associated Rectal Cancer

The literature on radiation-associated rectal cancer (RARC) remains inconclusive and often contradictory. This is partly due to the low prevalence of the disease, inconsistencies in defining the latency period between the use of radiation and the diagnosis of rectal adenocarcinoma, and variations in study design and methodology. Furthermore, results are affected by patient selection criteria and duration of follow-up. Several studies also analyzed RARC alongside other radiation-induced cancers, and so the unique characteristics of rectal cancer are often overlooked. Most studies reporting the incidence of RARC are based on large epidemiologic databases and tumor registries that contain large numbers of patients but often lack important details such as radiation dosages or radiation fields. Single-institution case series often provide treatment details but lack statistical power due to small sample size. Consequently, the biology, clinical characteristics, response to therapy, surgical and oncologic outcomes, and prognosis of RARC remain unknown.

## 2. Radiotherapy Is a Pillar of Prostate Cancer Treatment

Radiotherapy is an integral part of treatment for over half of all oncologic malignancies and has impacted the recurrence and survival rates of several cancers, including prostate cancer [1,2]. It is estimated that more than 260,000 new prostate cancers are diagnosed annually within the United States. It is the most common non-cutaneous cancer in males, with a steadily rising incidence in all age groups [3]. The implementation of screening programs contributes to an increasing number of patients diagnosed with prostate cancer at an earlier age, with 32% of new diagnoses occurring between the ages of 55 and 64 years [4].

The clinical behavior of prostate cancer is variable, ranging from indolent to aggressive [5]. There is a wide range of treatment options, including surveillance, androgen depletion therapy, immunotherapy, chemotherapy, radiotherapy, and surgery. Approximately 1 in 4 of all prostate cancer patients undergo radiotherapy [6], and the decision is based on patient stratification into risk groups. Factors such as age, life expectancy, TNM staging, PSA levels, the number of biopsy cores showing cancer, and Gleason grade are considered [7]. Radiotherapy is an accepted treatment modality for prostate cancer groups of all risks and all ages [7]. Radiotherapy is commonly utilized in older, more frail patients and those with comorbidities that preclude surgery. It is used less frequently in younger patients, who are more likely to undergo surgery and/or radiotherapy rather than active surveillance. Ultimately, many patients, both young and old, receive radiation for prostate cancer. A variety of therapeutic dosages are available and have been associated with decreased rates of biochemical failure, distant recurrence, and the need for salvage therapy [8]. However, the improvements in outcomes come at the cost of developing treatment-related toxicity, [9] including chronic genitourinary, sexual, and gastrointestinal dysfunction [10]. Patients may also develop additional primary cancers, such as RARC [11,12], which has been reported to develop in approximately 0.48% of prostate cancer patients treated with pelvic radiotherapy [13]. The 10-year overall survival of prostate cancer is over 80% [4,14,15]; therefore, many prostate cancer survivors who were treated with pelvic radiotherapy are at risk for developing radiation-associated second malignancies.

Many prostate cancer patients are treated with a combination of external beam radiotherapy (EBRT) and a brachytherapy boost, or with brachytherapy alone [7]. Brachytherapy involves the implantation of permanent radioactive seeds of either low-dose iodine-125 or palladium-103 for low-dose rate brachytherapy, or the placement of temporary iridium-192 catheters for high-dose treatment. Brachytherapy alone is supposed to be associated with less radiation exposure to the surrounding organs compared to EBRT [16]. However, brachytherapy has also been associated with the development of RARC. EBRT results in radiation exposure to the bladder neck, the anterior rectal wall, and the penile bulb. Patients treated with EBRT have been reported to be at a higher risk of RARC than patients treated with brachytherapy (Table 1) [17,18,19]. In summary, early detection of prostate cancer, a propensity to treat young patients in general, the increased use of radiation, and prolonged survival increase the risk for prostate cancer patients developing RARC.

## 3. Direct Beam Radiation Damages Surrounding Organs

Radiation-associated malignancies have been reported after treatment for breast, hematologic, gynecologic, and prostate cancers; however, the radiation dose which induces a malignancy remains unknown and the mechanism of carcinogenesis in the surrounding tissues has not been identified [21,30,31,32,33,34,35]. Ionizing radiation acts by targeting DNA, resulting in direct double-strand breaks, and by the production of free radicals, which indirectly induce DNA damage [1,36]. During treatment, patients are exposed to both primary and secondary radiation. Primary radiation occurs when cells are exposed to direct beam radiation. This includes tissue that is in the path of the photon beam and receives high doses of radiation. Scatter radiation (secondary radiation) occurs when photons spread out in different directions as it travels through the body. Ultimately, the surrounding tissues are exposed to both primary and secondary radiation, which may accumulate sublethal DNA damage in the cells [34]. Patients treated with EBRT of the prostate will have radiation exposure to the rectum in both of these ways.

## 4. The Incidence and Risk of Developing RARC

Both prostate and rectal cancer are common diagnoses. In the United States, the lifetime risk of developing prostate cancer is 11%, [37] and 0.65–1.2% for rectal cancer [38]. The lifetime risk of developing RCRPC remains unknown, and the literature appears to be conflicted in reporting associations between rectal cancer and prostate radiotherapy. Most studies investigating rectal cancer in patients who had undergone radiotherapy for prostate cancer (RCRPC)—compared to either rectal cancer in patients who had undergone treatment for prostate cancer without radiotherapy (RCNRPC) or rectal cancer in patients without a history of prostate cancer (RCNAPC)—are performed using population-based cancer registries, and the discrepancies in outcomes can be explained by differences in methodology. Most importantly, variations in latency period cutoffs (time from prostate irradiation to diagnosis of rectal cancer), patient follow-up, and radiation modalities can drastically affect the observations reported. Smaller studies are also limited by sample size. A summary of the studies can be found in Table 1.

Several studies found no association between rectal cancer and prostate radiotherapy. Neugut et al. performed a retrospective cohort study of 141,761 prostate cancer patients using the Surveillance, Epidemiology, and End Results (SEER) database. The authors found an increased risk of bladder cancer but did not observe a significant difference in the number of rectal cancer diagnoses in prostate cancer patients treated with or without radiation. The relative risks after eight years of follow-up were 0.8 (95% CI 0.4–1.3) for irradiated patients and 0.8 (95% CI 0.6–1.1) for non-irradiated patients [20]. Patients in this study were included if their latency period from prostate radiotherapy to rectal cancer diagnosis was at least six months. The duration of follow-up was not reported. Shortly after that study was published, Brenner et al. performed a similar analysis using the same study period in 122,123 patients from the SEER database [21]. This study included an additional four years of follow-up, and authors observed a 105% increased risk in RCRPC patients compared to RCNRPC patients. Juong et al. performed a similar analysis using the Korean Cancer Registry and did not find a difference in rectal cancer incidence in patients treated with prostate radiotherapy [28]. However, the authors included patients with a short latency period of at least two months and a median follow-up of only 3.5 years. Hinnen et al. used data retrieved from the Netherlands Cancer Registry to compare 1,187 patients who received only prostate brachytherapy to 701 patients treated with prostatectomy [27]. The authors found no difference in the incidence of RCRPC when compared to patients treated with prostatectomy. However, only 18 (<1.0%) patients were diagnosed with rectal cancer, limiting the power of the study. Similarly, Boorjian et al. identified 31 patients who developed rectal cancer in the Cancer of the Prostate Strategic Urologic Research Endeavor (CaPSURE) database [24]. Eleven (35%) of the patients had received radiation for their prostate cancer. The sample size was too small to measure an association between radiotherapy and rectal cancer (*p* = 0.14), and the latency period was too short to make such conclusions. Using the Geneva Cancer Registry, the standardized incidence ratio (SIR) of RCRPC was reported to be 5.26 (95% CI 0.2–29.3) after 10 years of follow-up in a study by Rapiti et al., despite using a latency period of at least 5 years and a median follow-up of 7.6 years. However, as only three RCRPC and three RCNRPC patients were identified after five years of follow-up, a longer follow-up of these patients is needed to accurately measure an association. These results highlight the importance of using appropriate methodology when investigating RARC, as the latency period and patient follow-up play a significant role in the outcomes of the studies.

The more recent literature addresses the issues of latency periods, patient follow-up, and sample size. Baxter et al. performed a population-based study and compared 30,552 prostate cancer patients treated with radiotherapy with 55,263 patients treated with prostatectomy. The authors found a 70% increased risk of developing rectal cancer [HR 1.70 (95% CI 1.4–2.2); *p* < 0.0001] in prostate cancer patients treated with radiation compared to those treated with surgery [22]. Patients with a latency period shorter than five years were excluded, and the mean length of follow-up was approximately nine years. Moon et al. performed a similar analysis using the SEER database and found that the odds of developing rectal cancer were higher in patients treated with prostate EBRT compared to non-irradiated patients [OR 1.60 (95% CI 1.29–1.99); *p* < 0.05]. The authors included patients with a latency period of at least five years, and median follow-up was 10 years [17]. Similar results have also been reported by Rombouts et al., using the Netherlands Cancer Registry; the incidence of rectal cancer in patients with prostate cancer treated with radiation therapy compared to patients treated without radiation was SIR 1.20 (95% CI 1.10–1.30) vs. 0.99 (95% CI 0.91–1.06), respectively [29]. Huo et al. highlighted the importance of appropriate patient follow-up in studies investigating radiation-induced malignancies. The authors performed a SEER study of 635,910 patients and found that the incidence of RCRPC did not differ from RCNRPC when the analysis included patients with a short latency period [SIR 1.06 vs. 0.92, *p* = 0.08]; however, they found a difference between groups when patients with a latency period of >10 years were compared [SIR 1.44 vs. 0.76, *p* < 0.0001] [26].

The risk of developing rectal cancer does not appear to increase in prostate cancer patients who underwent radical prostatectomy without radiotherapy [18,26,27]. For example, Yang et al. identified 1,491 RARC patients from 573,306 people treated with pelvic radiation for prostate, bladder, uterine, cervical, or ovarian cancer. At the end of follow-up, the authors reported that the incidence of RARC was highest in patients treated with a combination of EBRT and BT [SIR 1.85 (95% CI: 1.60–2.14)], followed by EBRT alone [SIR 1.22 (95% CI: 1.09–1.36)], and the risk was lower in patients treated without pelvic radiation SIR = 0.85, 95% CI: 0.80–0.91)] [35]. This observation suggests that radiation is independently playing a role in tumorigenesis of rectal cancer. Furthermore, there is no reported increased incidence of metachronous colon cancer in prostate cancer patients treated with or without radiation. Baxter et al. investigated the risk of colorectal cancer in a cohort of 30,552 prostate cancer patients treated with or without radiotherapy using the SEER database [22]. The authors stratified the risk of colorectal cancer into three anatomical groups: (A) definitely irradiated sites (the rectum); (B) potentially irradiated sites, such as the rectosigmoid colon, sigmoid colon, and cecum; and (C) non-irradiated large bowel, such as the transverse colon. The authors found a 70% increase in the risk of developing rectal cancer (HR 1.7, 95% CI 1.4–2.2) compared to patients who underwent radical prostatectomy only. There was no significant risk to the potentially irradiated and non-irradiated sites. Overall, there is strong evidence that the incidence of RARC appears to rise with time [39] and that RARC is a real concern and should be considered for young patients diagnosed with prostate cancer.

## 5. The Molecular Profile of RARC Remains Unknown

The genomic landscape of spontaneous colorectal cancer has been extensively characterized [40,41,42]. Colorectal cancers are associated with one of two forms of genetic instability, either chromosomal instability (CIN) or microsatellite instability (MSI), characterized by the accumulation of mutations in several oncogenes and tumor mutation genes [43,44]. However, the molecular profile of RARC remains unknown. Tsuji et al. analyzed the genetic aberrations of 5 RARC patients treated with pelvic radiotherapy for cervical cancer. None of the tumors had microsatellite instability based on standard enriched-PCR sequence analysis. Kras mutations were detected in two samples, and p53 in another two carcinomas. The authors concluded that RARC developed along the CIN pathway that characterizes the multistep dysplasia–adenoma–carcinoma sequence, rather than the mutator phenotype pathway [45,46].

While the information on the genomics of RARC is limited, differences in the molecular profiles of other radiation-associated malignancies and their non-irradiated counterparts have been reported. Shon et al. compared several molecular features in primary and radiation-associated cutaneous angiosarcomas in breast cancer patients using fluorescent in situ hybridization and immunohistochemistry. The authors observed gene amplification of *Myc* and its downstream effectors in all twenty of the radiation-associated angiosarcoma patients but not in any of the 18 primary angiosarcoma patients [47,48]. In adults, various *RET*/PTC rearrangements have also been described as the predominant driver mutation in radiation-induced papillary thyroid cancer, while BRAF V600E mutations are more common in sporadic papillary thyroid cancer [49]. In patients with radiation-induced thyroid cancer, the *RET*/PTC3 rearrangements are associated with a more aggressive phenotype, with advanced stage at diagnosis and poor prognosis [50]. Furthermore, Sha et al. performed whole-exome sequencing on 27 radiation-associated muscle-invasive bladder cancers and compared them to non-irradiated bladder cancers [51]. The authors reported a higher rate of mutations in DNA repair genes, such as FANCA (38.1% vs. 5.1%, *p* < 0.001), CHECK2 (26.1% vs. 2.5%, *p* < 0.001), and MSH6 (26.1% vs. 2.5%, *p* < 0.001) in radiation-associated muscle-invasive bladder cancer compared to the non-radiated bladder cancer group. Based on these observations, radiation-induced malignancies may have unique molecular features compared to their non-radiation-induced counterparts, but larger studies are needed.

## 6. Assessment, Surveillance and Diagnosis

Patients with RARC may be entirely asymptomatic and diagnosed incidentally during follow-up for the primary malignancy, or as a result of screening for colorectal cancer. Symptomatic patients most commonly present with overt or occult rectal bleeding resulting in iron deficiency anemia, hematochezia, discharge, urgency, diarrhea, and pain. However, these symptoms are also common in patients who develop radiation proctitis after pelvic radiotherapy [52]. Fibrosis may result in strictures, and patients may present with constipation or obstruction. A change in the gut microbiota may result in malabsorption [53]. Bulky lesions can cause changes in bowel habits, obstruction, or tenesmus.

The U.S. Preventive Services Task Force does not explicitly recognize prior pelvic irradiation as a risk factor for the development of rectal cancer [54]. The National Comprehensive Cancer Network recommends increased frequency of colorectal cancer screening in patients with previous abdominal, flank, and pelvic RT of ≥20 Gy [55]. Several authors support including history of pelvic irradiation as a risk factor for colorectal cancer and recommend more frequent surveillance compared to the average population. They also suggest a possible improvement in survival [19,22,56,57,58].

Endoscopy with biopsy remains the gold standard for the diagnosis of rectal cancer (Figure 1). Some patients may develop RARC in a background of chronic radiation-related changes, but this is not always the case [57]. When chronic proctitis is present, telangiectasia is the most common treatment-related change after EBRT to the prostate. It has been observed to frequently develop along the anterior distal rectal wall. Mucosal congestion, ulcerations, strictures, and necrosis have also been observed in these patients [59]. When radiation proctitis is present, diagnosing RARC becomes more difficult because smaller, ulcerated tumors are difficult to identify (Figure 2).

Like other rectal cancers, patients with RARC are staged using a combination of endoscopy, computed tomography (CT), and magnetic resonance imaging (MRI) or endoscopic ultrasound. The purpose of a CT scan is to evaluate the chest, abdomen, and pelvis for distant metastasis. Some patients undergo an endoscopic ultrasound of the rectum to assess the primary tumor (Figure 3A). Post-radiation changes in the rectal wall and surrounding structures may increase the difficulty of staging rectal cancer, [60] and the presence of radiation seeds may cause artifacts that can impair the interpretation of rectal MRI (Figure 3B). Ultimately, the accuracy of imaging studies for staging locoregional RARC remains unknown.

## 7. Treatment, Outcomes and Prognosis of RARC

Surgery is the primary treatment for rectal cancer. Early-stage spontaneous rectal cancer is often treated with surgery alone, either local excision or total mesorectal excision (TME). Radiotherapy delivered as either chemoradiation (45–50 Gy in 25–28 fractions combined with sensitizing fluoropyrimidines) or short-course radiation (25 Gy delivered in 5 fractions) is commonly indicated in patients with locally advanced rectal cancer to reduce the risk of local recurrence after TME surgery. However, radiotherapy is also commonly delivered before local excision in patients with early-stage rectal cancer who want to avoid the consequences of TME surgery [61]. The finding that patients with a complete response to neoadjuvant therapy can forgo TME surgery and achieve sustained organ preservation has emphasized the importance of radiotherapy in rectal cancer [62].

The use of radiotherapy in rectal cancer patients with a history of previous pelvic radiation is controversial. The idea of repeating radiation therapy of a potential RARC seems counterintuitive. The benefits of reirradiation for local tumor control in patients with either early-stage or locally advanced RARC have not been proven. Furthermore, pelvic reirradiation is associated with the risk of severe toxicities such as severe radiation proctitis, genitourinary bleeding, bone fractures, bowel, urinary, and sexual dysfunction, and pelvic pain [63,64]. Reirradiation followed by local excision could lead to delayed healing of the rectal wall wound and rectourethral or rectovaginal fistula [65]. There are also concerns that the pre-existing radiation-related fibrosis could be enhanced by reirradiation, further increasing the technical difficulty of the TME operation, and impacting its quality. Reirradiation may increase the anastomotic leak rate in patients who undergo a sphincter-saving procedure and the rate of perineal wound complications in patients requiring an abdominoperineal excision of the rectum [66,67,68]. These considerations highlight the complexity of selecting appropriate patients for neoadjuvant rectal radiation. Systemic chemotherapy is probably the neoadjuvant therapy of choice in patients with locally advanced RARC. However, accelerated hyper-fractionated radiation with concomitant capecitabine remains an option for RARC patients with an incomplete response to systemic chemotherapy [69,70,71,72].

Surgery for RARC patients also deserves special technical considerations. Most of the tumors are located in the distal rectum, often in the anterior wall adjacent to the previously irradiated prostate. Given the absence of the mesorectum in the anterior wall of the distal rectum, tumors usually involve or threaten the circumferential resection margin, [73] often requiring neoadjuvant therapy, extended surgery, or both. Considering the limited neoadjuvant options for previously irradiated patients and the impaired healing of heavily irradiated prostatic and urethral tissue, an extended resection often requires pelvic exenteration, a procedure associated with significant mortality and morbidity rates in older patients with comorbid conditions. Patients diagnosed with RARC who cannot undergo surgery require alternative treatment approaches. Some examples include chemotherapy, tumor ablation, pelvic reirradiation using high dose endorectal brachytherapy, and diverting stomas; however, the literature supporting the use of these alternative approaches is limited.

## 8. The Oncologic Outcomes of RARC

In 2019, Rombouts et al. used data retrieved from the Netherlands Cancer Registry in order to compare the five-year overall survival of 618 rectal cancer patients previously treated with pelvic radiation to 750 patients treated without radiotherapy [29]. The authors found no differences in survival between the two groups (HR 0.94, 95% CI 0.79–1.11). Yang et al. published SEER data which compared the five-year survival of patients with RARC to primary rectal cancer without a previous pelvic cancer or pelvic radiotherapy (called the “PRCO” group). The authors performed propensity score matching of the two groups based on gender, age, race, stage, and treatment with chemotherapy, radiation, and surgery. In this study, the survival of rectal cancer patients treated with radiation and surgery were not directly compared. However, RARC was associated with lower five-year overall survival than the matched PRCO group (*p* < 0.001, HR 1.33 (95% CI 1.14–1.55). Furthermore, the RARC group was also found to have shorter rectal cancer-specific survival (*p* = 0.004, HR 1.30, 95% CI 1.07–1.58) compared to their matched, non-irradiated counterparts. However, a difference in rectal cancer-specific survival (RCSS) was not observed between patients treated without pelvic radiation and matched patients without pelvic cancer (*p* = 0.116; HR 1.11, 95% CI 0.97–1.28), suggesting that prognosis is worse in patients previously treated with pelvic radiotherapy [19]. Both groups were well matched, implying that the lower survival of the RARC group may be attributed to differences in tumor biology and the use of radiation. Similar analyses have been attempted in female patients diagnosed with rectal cancer after radiotherapy for cervical cancer, but they are limited by sample size [74]. Hung et al. identified 45 patients with rectal cancer after radiotherapy of the cervix. The authors reported a five-year overall survival rate of 28.7% in the 29 patients who underwent previous cervical radiotherapy compared to 67.2% in the non-irradiated cohort (*p =* 0.081). Despite such efforts, local regrowth and recurrence rates, distant recurrence rates, and disease-free survival have not yet been reported in the literature.

## 9. Risk Reduction and Prevention

Preventative measures to reduce the exposure of the surrounding organs to radiation may help to reduce the incidence of RARC. These include minimizing exposure based on radiotherapy modality, and individualized dose adjustments using dose–volume histograms [75]. Rectal spacers can be used to temporarily displace the rectum posteriorly, reducing the dosage of radiation absorbed by the rectum [76,77,78]. Other radiotherapy modalities to treat prostate cancer should also be considered. For example, proton-beam radiation therapy has been shown to be associated with a lower risk of second cancer compared to intensity-modulated radiotherapy (OR 0.31, 95% CI 0.26–0.36, *p* < 0.0001) [79]. Furthermore, bone marrow irradiation or bone-sparing pelvic radiotherapy may be utilized, although its use in patients with RARC and effects on their immune system have yet to be studied. Patients with germline mutations at risk for additional cancers should be identified early in their clinical course, and follow-ups should be frequent [80]. Smokers should be advised to quit, diet modifications should be encouraged, and patients should be educated on the possibility of developing another cancer.

## 10. Future Directions

There are several of areas of research that need to be addressed for patients with RARC. First, any unique clinical and treatment characteristics which impact survival must be identified. Additionally, a more comprehensive understanding of their oncologic outcomes will provide insight into the tumor biology and prognosis of RARC. Furthermore, performing DNA and RNA sequencing of tumor samples from RCRPC may shed light on the potential differences in mutational signatures and cell-intrinsic transcriptional programs compared to patients with RCNAPC. Finally, a better understanding of the complex interactions between the host immune system and the tumor evolution of RARC may provide physicians with a better understanding of the disease and an opportunity to personalize its treatment.

## 11. Conclusions

There is a high prevalence of patients with prostate cancer treated with radiation therapy worldwide. These patients are at a higher risk of developing rectal cancer, but the reported incidence of RARC is largely dependent on the latency period and length of follow-up. Patients diagnosed with RARC appear to have worse outcomes compared to patients with rectal cancer without a history of pelvic irradiation; however, the molecular drivers of pathogenesis, clinical characteristics of this disease, and optimal treatment methods remain unreported. These findings suggest that RARC is a real concern for patients and challenging to manage for clinicians. A better understanding of this disease may provide relevant insights into the unique attributes of RARC and may ultimately help to improve prognostication and treatment for this unique population.

## Figures and Tables

**Figure 1 cancers-15-02214-f001:**
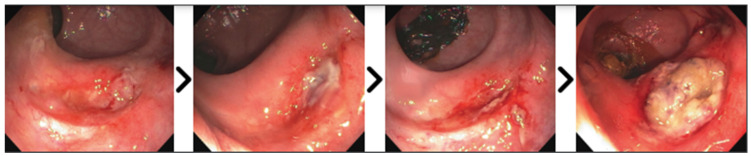
Interval endoscopies of a patient who refused treatment for RARC with disease progression. Over a period of one year, the tumor evolved from an ulcer along the anterior rectal wall.

**Figure 2 cancers-15-02214-f002:**
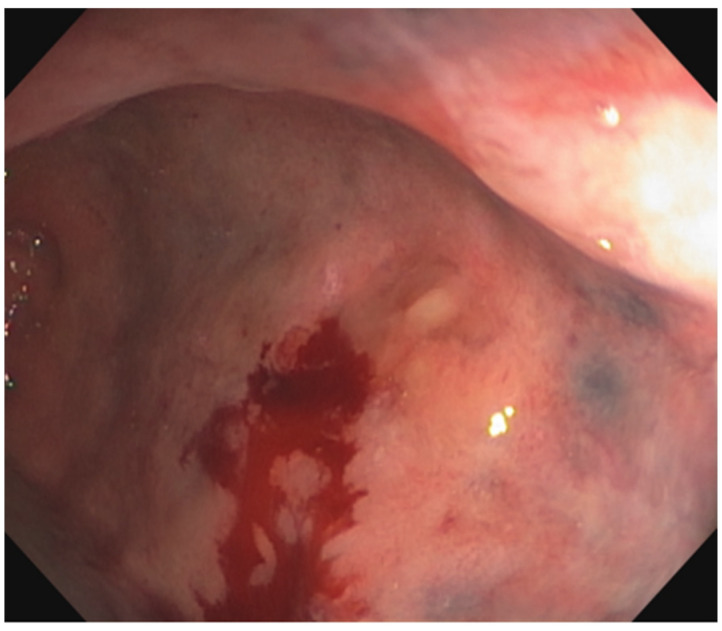
Endoscopy from a patient whose lesion was arising from a background of rectal fibrosis. A biopsy of this lesion demonstrated fragments of adenocarcinoma.

**Figure 3 cancers-15-02214-f003:**
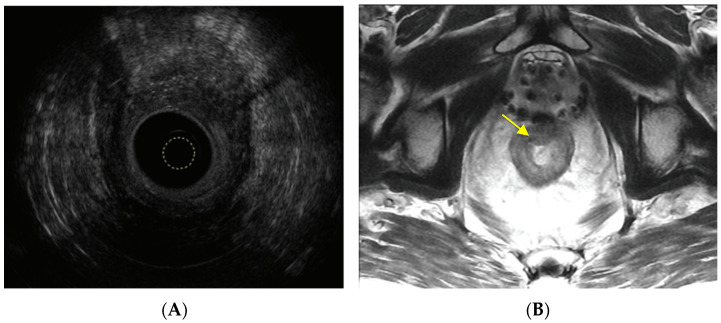
(**A**) Endoscopic ultrasound of a uT3N0Mx radiation-associated rectal cancer along the anterior rectal wall. The yellow circle marks the tip of the ultrasound probe. (**B**) MRI of the rectum depicting a radiation-associated rectal cancer in a patient who underwent brachytherapy for prostate cancer. The yellow arrow marks the tumor.

**Table 1 cancers-15-02214-t001:** A list of the literature reporting the incidence, risk, and survival of radiation-associated rectal cancer compared to non-irradiated patients [located below citations].

Study	Data Source	Cohort Size *	Latency Period Inclusion	Follow-Up Duration	Radiation Modality	Control Group	Analysis	Findings **
Neugut et al. (1997) [20]	SEER	141,761	≥6 mo	Not reported †	Not reported	RT−	Time from treatment	
							6 mo < 5 yrs	RT+: RR 0.7 (0.5–0.9) RT−: RR 0.8 (0.7–0.9)
							5–8 yrs	RT+: RR 0.8 (0.5–1.2) RT−: RR 0.8 (0.6–1.0)
							>8 yrs	RT+: RR 0.8 (0.4–1.3) RT−: RR 0.8 (0.6–1.1)
Brenner et al. (2000) [21]	SEER	122,123	≥2 mo	≥10 yrs	Not reported	Sx	Time from treatment	
							≥5 yrs	35% increased risk (95% CI, −1, 86); *p* = 0.06
							≥10 yrs	105% increased risk (95% CI, 9, 292); *p* = 0.03
Baxter et al. (2005) [22]	SEER	85,815	≥5 yrs	Mean >9 yrs	EBRT	Sx	Definitely irradiated sites Potentially irradiated sites Non-irradiated sites	HR 1.70 (95% CI: 1.4–2.2); *p* < 0.0001 HR 1.08 (95% CI: 0.92–1.26); *p* = 0.35 HR 0.95 (95% CI: 0.78–1.15); *p* = 0.58
								RT+ vs. RT−: 70% increase in the development of RARC
Neider et al. (2005) [18]	SEER	243,082	≥6 mo	>10 yrs	EBRT BT EBRT-BT	Sx	Time from treatment Entire follow-up period	EBRT: HR 1.26 (95% CI: 1.08, 1.47) BT: HR 1.08 (95% CI: 0.77, 1.54) EBRT−BT: HR 1.21 (95% CI: 0.89, 1.65)
								Risks vary depending on RT modality and time
Kendal et al. (2006) [23]	SEER	237,773	≥5 yrs	Median 5 yrs	EBRT	Sx WW	Time from treatment Entire follow-up period	RT+ vs. Sx: RR 2.38 (95% CI: 2.21–2.55) Sx vs. WW: RR 3.44 (95% CI: 3.22–3.67)
								The WW group had a higher risk of rectal cancer than RT+ group, suggesting radiation does not influence this process.
Moon et al. (2006) [17]	SEER	140,767	≥5 yrs	Median 10 yrs	EBRT	RT−	Time from treatment ≥5 yrs	EBRT: OR 1.60 (95% CI, 1.29–1.99)
								BT, either in isolation or in combination with EBRT, did not show significantly different odds of RARC
Boorjian et al. (2007) [24]	CaPSURE	9681	≥30 days	Median 39 mo	EBRT BT EBRT-RT	RT−	Entire follow-up period	RT+: 11/31 patients (35%, *p* = 0.14)
Rapiti et al. (2008) [25]	GCR	1134	≥5 yrs	Median 7.4 yrs	EBRT	RT−	Time from treatment	
							5–9 yrs	RT+: SIR 1.2 (95% CI: 0.04–6.9) RT−: SIR 1.5 (95% CI: 0.4–3.9)
							≥10 yrs	RT+: SIR 5.3 (95% CI: 0.2–29.3) RT−: undefined
								Increased incidence of colon cancer in 13 patients
Huo et al. (2009) [26]	SEER	635,910	Notreported	>10 yrs	EBRT BT	RT−	Time from treatment	
							<6 mo (*p* = 0.02)	RT+: SIR 0.99 (95% CI: 0.77–1.27) RT−: SIR 1.38 (95% CI: 1.19–1.60)
							6 mo–5 yrs (*p* = 0.95)	RT+: SIR 0.96 (95% CI: 0.88–1.05) RT−: SIR 0.96 (95% CI: 0.90–1.02)
							>5–10 yrs (*p* = 0.08)	RT+: SIR 1.06 (95% CI: 0.93–1.20) RT−: SIR 0.92 (95% CI: 0.84–1.01)
							>10 yrs (*p* < 0.0001)	RT+: SIR 1.44 (95% CI: 1.22–1.71) RT−: SIR 0.76 (95% CI: 0.65–0.88)
							Entire follow-up period (*p* = 0.03)	RT+: RR 1.04 (95% CI: 0.97–1.11) RT−: RR 0.95 (95% CI: 0.91–1.00)
Hinnen et al. (2011) [27]	NCR	1888	≥1 yr	Median 7.5 yrs	BT	Sx	Time from treatment	
							1–4 yrs	BT: RR 0.41 (95% CI: 0.05–1.48) RP: RR 2.12 (95% CI: 0.69–4.94)
							5–15 yrs	BT: RR 1.78 (95% CI: 0.71–3.67) RP: RR 0.96 (95% CI:0.19–2.81)
Joung et al. (2015) [28]	KCR	55,378	>2 mo	Median 3.5 yrs	Not reported	RT−	Time from treatment	RT+: SIR 1.03 (CI not reported) RT−: SIR 0.67 (CI not reported), *p* < 0.05
Rombouts et al. (2020) [29]	NCR	96,577	Not reported	Median 6.8 yrs	Not reported	RT−	Time from treatment	RT+: SIR 1.20 (95% CI: 1.10–1.30) RT−: SIR 0.99 (95% CI: 0.91–1.06)
							Subhazard ratio Crude incidence ratio	SHR: 1.89 (95% CI: 1.66–2.16) CIR: 1.3% RT+ vs. 0.7% RT− (*p* < 0.001)
							5-yr overall survival	RT+: 33.7% (95% CI: 29.6–37.8) RT−: 39.1% (95% CI: 35.4–42.8)
Yang et al. (2020) [19]	SEER	291,395	≤5 yrs	>20 yrs	EBRT BT EBRT-BT	RT−	Time from treatment	
							Entire follow-up period	RT+, EBRT: SIR 1.22 (95% CI: 1.09–1.36) RT+, EBRT−BT: SIR 1.85 (95% CI: 1.60–2.14) RT−: SIR 0.85 (95% CI: 0.8–0.91)
							5-yr OS	Significantly lower in RT−SPRC vs. matched-PRCO group HR = 1.33 (95% CI: 1.14–1.55); *p* < 0.001
							5-yr RCSS	Significantly lower in RT−SPRC vs. matched-PRCO group HR = 1.30 (95% CI: 1.07–1.58); *p* = 0.004

* Cohort size. ** Risks, incidences and outcomes of rectal cancer. † Not reported: not explicitly stated by the authors. Sx: treated with surgery. RT+: treated with radiation. RT−: treated without radiation. WW: prostate cancer group undergoing active surveillance. OS: overall survival. RCSS: rectal cancer-specific survival. SEER: Surveillance, Epidemiology, and End Results Cancer Registry. NCR: Netherlands Cancer Registry. CaPSURE: Cancer of the Prostate Strategic Urologic Research Endeavor. GCR: Geneva Cancer Registry. KCR: Korean Cancer Registry. RT-SPRC: second primary rectal cancer after radiotherapy (≈RARC). PRCO: primary rectal cancer only (≈rectal cancer without previous pelvic cancer). EBRT: external beam radiation therapy. BT: brachytherapy. RP: radical prostatectomy. HR: hazard ratio. OR: odds ratio. CI: confidence interval. Mo: months. yr: year. yrs: years.

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
