# Peer review of "Rectal Cancer after Prostate Radiation: A Complex and Controversial Disease"

_cancers, 2023, doi:10.3390/cancers15082214_

Round 1
Reviewer 1 Report
In the review "Rectal Cancer after Prostate Radiation: A Complex and Controversial Disease" Omer et al. addresses an often neglected subject, a 2nd malignancy induced by radiotherapy and the possible causal relationship between prostate cancer radiotherapy and radiation-associated rectal cancer (RARC). The authors mention the limits of the current studies, the effects of radiotherapy on OARs and also they analyze all the important aspects of RARC including the incidence, risk, diagnosis, molecular profile, surveillance, treatment and outcomes and prognosis, but also the risk reduction strategies and prevention methods RARC. The article also includes a table with comparative incidence and survival data of RARC and endoscopy/ultrasound/MRI images including post-radiotherapy lesions (fibrosis), a possible starting point for adenocarcinoma (RARC). Mentioning fibrosis, I would also like to highlight the evolution of doses and treatment techniques in prostate cancer up to a very high escalation (proposed by Zelefsky et al. - doi: 10.1016/j.ijrobp.2011.11.047.), but I would also mention hypofractionated regimens possibly associated with a risk of late fibrosis. Also, as a radiation oncologist, I would mention an important aspect, not for the incidence, but for a possible unfavorable evolution of RARC related to immune host status: bone marrow irradiation and bone sparing radiotherapy as a strategy). However, these topics seem to be the subject of future studies, so I recommend publishing the article in this form
Reviewer 2 Report
This is a Novel and comprehensive presentation of published information regarding Radiation Associated Rectal Cancer developing in men following radiation therapy in the management of Prostate Cancer. The tendered conclusions are appropriate as definitive findings are not available. This is due to the fact that these tumor's development and treatment are quite delayed and infrequent. The authors have performed an extensive and comprehensive review of the literature and given the heterogeneity of the disease the outcomes, and the conclusions are clear, not over or understated, and definitely supported by the evidence.
Reviewer 3 Report
This paper is very excellent and can be acceptable.
Reviewer 4 Report
Dear authors,
This is a well-written valuable review of literatures about rectal cancer associated with radiotherapy for prostate cancer. The readers certainly will be interested in this article. I think this is worth publishing.
Prostate cancer is one of the most common malignancies, which is frequently treated with irradiation. Since patients with prostate cancer has relatively good prognosis and rectal cancer is a common malignancy, the likelihood the physician encounter patients with RARC (radiation-associated rectal cancer) is high.
Young patient with prostate cancer is preferably treated with surgery or brachytherapy rather than EBRT. If patients with prostate cancer underwent EBRT, surveillance for rectal cancer would be necessary especially after 5 or 10 years. The management of RARC need special consideration because of high complication rate and poor prognosis.
I have some minor comments.
In page 6, image 1 (endoscopic figures) cannot be shown completely on the PDF file because the right part was cut.
In page 8 line 323, “RCSS” means “rectal cancer specific survival”? Would you confirm that?
Regards,
Reviewer 5 Report
The paper has a logical consecutive structure, which helps the reader gain inside step by step. It progressively gives an appropraite overview of RARC concerning incidence, diagnosis, treatment, prognosis etc.
However, in the end the group could have spend some more time on risk prevention and reduction, going further into different techniques to minimalize the risk of RARC?
The abstract gives a good overview and introduction but is missing a sentence of conclusion. For a better understanding, maybe add a one-sentence Take-Home-Message?
The paper hints towards the necessity for longer trials when evaluating the risk of RARC, however this is not mentioned in the conclusion. The paper extensively compares study designs and findings and highlights the link between latency period and follow-up and study outcome. In Line 190-192 the authors suggest that RARC is a strong concern however this thought is not again mentioned in the conlcusion. The authors seem to sway towards the opinion that RARC is indeed induced by prostate radiotherapy but that is not mentioned in summary or conclusion.
Depending on personal preferences, the group might also consider adding some graphs (e.g. Kaplan Meier curves) of the mentioned trials, giving the reader a more graphic comparison of the presented data.
the paper tries to examine the molecular profile of RARC. Here it takes a side step and talks about mutations in radiation-associated angiosarcoma in breast cancer and radiation-induced thyroid cancer. This would hold up as a good comparison and example of differences in molecular profiles when talking about radiation induced tumors, however the lack of data on RARC mentioned in the end of the passage somehow relativizes these findigs in this context. Could the group add some data here or is there nothing available? If no data is present, then the authors might consider shortening the paragraph a little. While it demonstrates the individual molecular profiles of radiation induced carcinomas, it might confuse some readers because of a missing link or rather lack of data and slightely takes away the focus from the topic at hand.
In line 246ff the authors talk about difficulties diagnosing RARC simultaneously with radiation proctitis. I think it would be interesting to mention in one or two sentences, whether a comorbidity of proctitis is linked to lower survival of RARC due to a later diagnosis? Is there some data available?
Language issues:
Spelling mistake in line 357?: first ‚of‘ in first sentence of the parapraph ‚future directions‘ too much?
Overall, the group presents a high-quality literature review, highlighting pros and cons of different study approaches and helps recognizing RARC as a serious treatment complication after radiotherapy of prostate cancer.
I would suggest accepting the paper, the authors might think about some adjustments as mentioned above, but the paper is indeed a well done review.
